# The Promise and Challenge of High Pressure Macromolecular Crystallography

Katarzyna Kurpiewska [1],*, Joanna Sławek [2],*, Agnieszka Klonecka [2,3,4] and Maciej Kozak [2,5]

1 Department of Crystal Chemistry and Crystal Physics, Faculty of Chemistry, Jagiellonian University, 30-387 Kraków, Poland

2 SOLARIS National Synchrotron Radiation Centre, Jagiellonian University, 30-392 Kraków, Poland

3 Doctoral School of Exact and Natural Sciences, Jagiellonian University, 30-348 Kraków, Poland

4 The Faculty of Physics, Astronomy and Applied Computer Science, Jagiellonian University, 30-348 Kraków, Poland

5 Department of Biomedical Physics, Faculty of Physics, Adam Mickiewicz University, 61-614 Poznań, Poland

* Correspondence: katarzyna.kurpiewska@uj.edu.pl (K.K.); joanna.slawek@uj.edu.pl (J.S.); Tel.: +48-12-686-24-66 (K.K.); +48-12-664-41-14 (J.S.)

**Abstract:** Since its introduction in the early 1970s, high pressure crystallography (HPX) has shown great potential for the investigation of different types of matter. Using diamond anvil cells, HPX is an emerging technique that has been rapidly implemented, making it available to biologists, and there is immense potential for utilizing this technique in biological systems in the future. At the molecular level, high-pressure crystallographic investigation provides information on structural characteristics that not only determine the native conformation of a protein but also the conformations with higher free-energy, thus revealing function-related structural changes and properties that can be modified as a result of pressurization. The increase in the number of crystal structures of different macromolecules determined under high pressure over the last five decades can be ascribed mainly to two factors: the emergence of high-pressure cells with very large, open angles, and the advent of third generation synchrotron sources. The use of high pressure crystallography as a research tool has been shown to contribute to the advancements in the basic fields of biochemistry (protein misfolding and aggregation), biophysics (protein stability), and biotechnology (food processing). Presently, with a growing interest in biomedicine and nanotechnology, this nonstandard method appears to be a valid instrument for probing more challenging and complex systems. In this review, we present the method, highlight a selection of recent applications, and describe challenges for high pressure macromolecular crystallography (HPMX).

**Keywords:** high pressure macromolecular crystallography; protein structure; unfolding; crystal; X-ray analysis

## 1. Introduction

Most of our biochemical knowledge has been gained from studies carried out at or near an atmosphere pressure while much of the matter in the Universe exists under higher pressure (HP). Even the relatively small diversity of pressure on Earth determines the distribution of life. It is, therefore, critical to understand the fundamental attributes of biological systems not only based on what we know about life on Earth and within the limits from 0.1 to 110 MPa (the environment in the deepest ocean trenches, such as at the bottom of the Mariana Trench), but also to investigate how living matter behaves beyond this range. Although high pressure studies of biomacromolecules in solution starting from the early 1960s have become a standard method, the question of what happens on the structural level upon pressurization remained unknown for a long time. Doubts about the interest of high-pressure research in the crystal state connect with the findings revealing

that it is difficult to quantify the boundaries of life's ability to survive under extreme compression. Although various hypotheses exist regarding the modification strategies used by biological systems to withstand extreme pressure, the basic biophysical or biochemical mechanisms for providing protection are not yet fully comprehended. Furthermore, a set of pressures above which organisms and constituent macromolecules can no longer exist in a functional state turned out to depend on many factors. First, we should remember that the most essential biological component of cells is water. Taking into account that pure $H_2O$ crystallizes at approximately 1 GPa (producing the dense ice-VI phase) [1], understanding the physical and chemical properties of water under extreme compression and the way water influences the structure and function of biomolecules after pressurization is crucial. In the case of protein crystals closed inside a high-pressure cell, water contains dissolved ions (in general, pressure medium is largely composed of the crystallization mother liquor), thus even experiments beyond 1 GPa are possible. Nevertheless, the choice of the hydrostatic medium is of the highest importance and should be adapted to the properties of the sample (details in Section 3). Second, from a chemical point of view, macromolecules are by far the most structurally complex and functionally sophisticated molecules known, having diverse levels of structure and showing many different sizes and shapes. Compared with other condensed materials studied under high pressure, proteins definitely serve as a unique substance. For example, the observation that proteins in solution undergo denaturation in a comfortably achievable pressure range, and at the same time pressure effects on proteins in a crystal state are first visible above even a few times higher range, is a phenomenon [2]. In terms of stability, proteins in solution generally unfold at room temperature under 400–600 MPa, although very sensitive proteins can unfold at lower pressures (for example, the apple allergen Mal d1 unfolds at 200 MPa) [3]. Proteins with the highest resistance are capable of maintaining their native structure at pressures near or exceeding 1 GPa. For instance, *Methanococcus jannaschii*'s heat shock protein (HSP16.5), which originates from thermophilic archaea, undergoes denaturation at a pressure of 1.7 GPa [4], while milk proteins alkaline phosphatase and lactoperoxidase preserve their activity up to 800 MPa [5,6]. Nevertheless, pressure resistance of proteins cannot be predicted, and different pressure levels cause either local or global conformational changes in protein structure, which eventually lead to destabilization, denaturation, and aggregation. Another important property that controls the way the protein responds to compression is flexibility. This feature is especially essential for enzymes and their activity, which can be modulated by the introduction of high pressure. In fact, it is enzymology that nowadays benefits spectacularly from the high pressure studies [7]. Pressurization is used to stabilize enzymes and regulate both their catalytic property and specificity [8]. Moreover, considering the great potential of modern molecular biology in designing an amino acid sequence that via rational engineering or directed evolution leads to the development of enzymes with functions useful in biotechnology and medicine, a revolution in high pressure enzymology is observed. Such advanced procedures result in novel biocatalysts that fold into a specified three-dimensional structure and can withstand extreme conditions [9]. In recent decades, scientists have demonstrated that artificial enzymes can serve as inexpensive and extremely stable substitutes for natural enzymes in various applications [10]. Consequently, utilizing high-pressure protein engineering could facilitate enzyme-catalyzed synthesis of fine chemicals and pharmaceuticals, as well as the production of modified proteins with medical or pharmaceutical significance. As previously stated, it is understood that high hydrostatic pressure can trigger alterations in protein conformation and aggregation, thus studying protein structures under high pressure also holds medical significance [11]. High hydrostatic pressure is a powerful tool for investigating protein folding and the structure of intermediate folding states, which are recognized as key contributors to the protein misfolding process, and for studying protein folding and the structure of folding intermediates, which are known to play a pivotal role in the protein misfolding process. Exploring misfolded proteins, aggregates, and amyloids that originate from partially folded intermediates offers the potential for gaining a deeper comprehension of protein misfolding disorders [12]. The

primary benefit of such investigations is that high hydrostatic pressures discourage the hydrophobic and electrostatic interactions that lead to protein aggregation. Consequently, studying proteins under high hydrostatic pressures permits reversible formation of protein aggregates, facilitating detailed analysis of their structural, thermodynamic, and kinetic properties [13]. The facts concerning the stability and functioning of key biomolecular components under high pressure play an essential role in both understanding protein aggregation and reversing, not only in medicine but also in industrial applications [14]. Pressure ranges below 2 GPa can induce alterations in secondary to quaternary levels of protein structure, which regulate assembly pathways [15] and reaction equilibria [16]. It is generally believed that pressure can also be implemented to change the properties of entire cells and microorganisms. The increased immunogenicity observed in certain pressure-treated bacteria and viruses has already been utilized in the development of new vaccines [17]. In addition, the entire tumor cells inactivated with pressure can be used to trigger the effective antitumor immunity [18]. Furthermore, high pressure has the potential to be used for sub-zero temperature storage without freezing, for various biological specimens such as cells, animal tissues, blood cells, and organs for transplantation.

The barosensitivity of some microorganisms, including some of the most virulent foodborne pathogens, makes it possible to employ high-pressure processing in food production [19]. For example, pascalization is particularly useful with temperature-sensitive food products. High pressure produces extendedshelflife food products (with the characteristics of being fresh and preservative-free with original nutritional and organoleptic qualities) that have the necessary microbial stability [20]. Eliminating biological agents is anticipated to be feasible with pascalization (sterilization) for delicate biopharmaceuticals or medicinal compounds. It should also be mentioned that advancements in industrial methodologies [21] have significantly expanded the use of high pressure in recent years, both directly through mechanical disruption and indirectly as a result of the techniques employed [22]. Examples include cell lysis, analytical ultracentrifugation, high-performance liquid chromatography, high-pressure refolding of inclusion bodies [23], and integration of membrane proteins into nanodiscs [24].

In addition to providing a robust way to investigate the biotechnological aspects of compression, studies of biomolecules under extreme conditions help to explain the adaptation of organisms to changing environments. Many microbial communities have evolved to exist and endure extreme environmental conditions, including high hydrostatic pressure. HP is an essential tool in examining the trade-off between structural flexibility and stability in piezophiles, and how this balance is maintained for survival in extreme environments [25]. Looking beyond our planet and into the future, when searching for habitable environments elsewhere in the cosmos, low or high pressure must also be considered as a factor that shapes the habitability of other planets. It is now believed that pressure plays an important role not only in the distribution of life on Earth, but also might be a physical parameter that limits the life in the Universe.

In summary, given the importance of proteins in understanding the mechanisms of the functioning of living organisms on the one hand and the importance of the investigation of macromolecules under non-standard conditions on the other hand, the modern bioscience persistently seeks to develop unique HP methods. Pressure perturbation is used in combination with a wide range of techniques including circular dichroism [26], fluorescence [27], FT-IR spectroscopy [28], small-angle X-ray scattering (SAXS) [29], small-angle neutron scattering (SANS) [30], or NMR [31] and described in this review crystallography. Compared to spectroscopic techniques, X-ray crystallography is generally considered superior as it enables the straightforward determination of three-dimensional structures of proteins and their surrounding water structures. Therefore, macromolecular crystallography in combination with high pressure (HPMX), as a younger sister of conventional biocrystallography, allows the investigation of the structure and the explanation of the properties of macromolecules associated with all the numerous applications we have mentioned above, especially in terms of the conformation and functionality of the biomolecules (Figure 1).

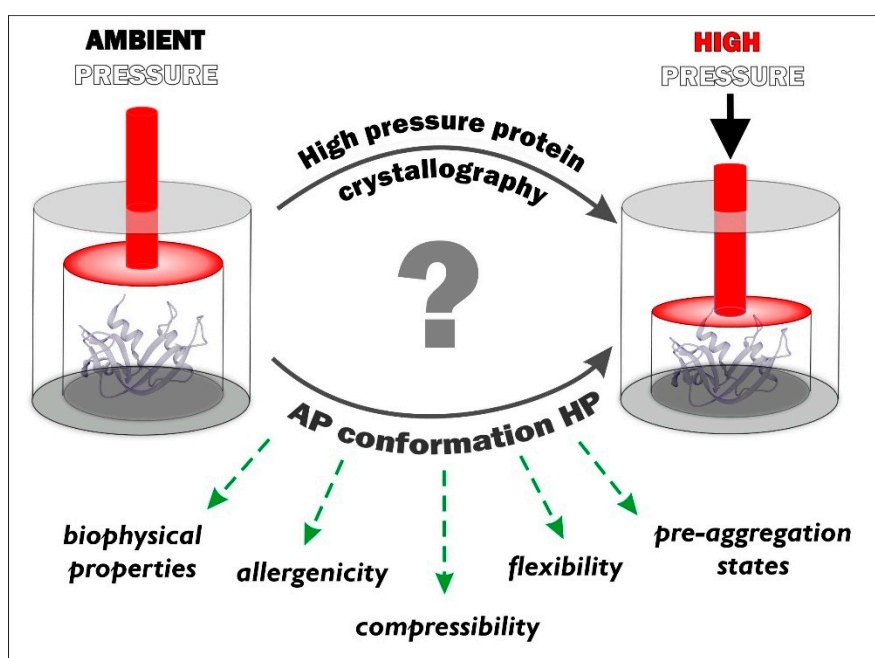

**Figure 1.** High pressure protein crystallography as a tool for investigation of macromolecular properties associated with high pressure conformation.

Although HPMX has some limitations, the results of the studies conducted so far are encouraging and have been shown to complete the picture of macromolecules investigated under HP in solution [32]. In static compression experiments on crystal structures, in addition to the challenges of handling delicate and temperature-sensitive crystals that are prone to over-drying when exposed to air, several other non-trivial aspects need to be addressed, which will be further discussed below. Although it is simple to imagine the compression of the crystals built from inorganic or organic molecules, it should be explained how the pressure is distributed into the protein crystal? This question is fundamental and therefore will be addressed before we introduce further aspects of HPMX experiment. The hydrostatic compression of the polypeptide chain is facilitated by the communication between the liquid surrounding the crystal and the liquid phase within the crystal, which occurs through channels present in the crystal structure. Compression would not be possible without the heart of the HPMX experiment—a diamond anvil cell (DAC). The latest design of the DAC has undergone significant improvements over the years and now features a widened physical angle from 30 to 120 degrees on both sides, which makes it an ideal tool for single crystal X-ray diffraction. This advancement is particularly noteworthy as it allows for access to a vast reciprocal space that is unmatched when conducting measurements under high hydrostatic pressure [33]. In this review, besides providing a general overview of the crystallographic experiment and selected examples of structural studies under high pressure without pretending to offer a complete set of HPMX achievements, we will also discuss aspects which appear to us crucial for future research in the field.

## 2. High Pressure in Structural Studies

The origins of high-pressure structural studies can be traced back to geology and astronomy, as a significant portion of condensed matter in the universe exists under high pressure conditions. For instance, a classic study could involve examining the reaction of a mineral structure to increasing pressure levels, up to those present at the corresponding depth within the Earth. As evidenced in previous HPX studies, the structures obtained at various pressure levels are analyzed to reveal the molecular responses to compression. The application of a few tens of MPa may only affect the weak intermolecular interactions, leading to a decrease in the unit cell volume and a possible rearrangement of structural

units. Higher pressures may result in changes in the conformation or internal geometry of macromolecules and ligands. At higher compression, notable phenomena include chemical reactions such as polymerization and transformations such as the generation of new polymorphs or reorganization of hydrogen bonding networks in organic and inorganic compounds. Additionally, there may be changes in molecular organization or conformation, closer approach of structural units, and even modifications to the electron configuration [34]. More recently, the scope of materials studied under high pressure has expanded to include biologically relevant molecules spanning from amino acids and DNA fragments to proteins. One significant advantage of applying high pressure in protein studies is the ability to explore intermediate substates between the folded and unfolded states. By combining experimental and theoretical approaches, it has been possible to map the energy landscape and construct p-T phase diagrams for several proteins [31].

Although crystal structures determined under HP comprise less than 0.1% of structures reported in the Protein Data Bank [35], studies at high pressures are becoming more common. The development of apparatus (compact high-pressure cells, high-intensity radiation sources and area detector diffractometers) followed by the improvement of technique (crystal mounting, pressure measurement, and data acquisition) over the past 10 years is remarkable, and HP studies are now conducted for more complex systems. When pressure is applied, different responses can be observed from the macromolecules that form the crystal and the crystal itself (Figure 2). The most important features reported as those that change in the course of the biocrystallographic HP experiment are changes of crystals quality [36,37], crystal symmetry [38], volume of the unit cell [39], and Wilson B factors when crystallographic parameters are concerned. In terms of modification at the molecular level, changes can be observed in the hydration structure [40], the size and localization of cavities and tunnels [41], and the conformation of the polypeptide chain and ligands [42]. All structural rearrangements are often associated with alterations in the stability and/or activity of biomolecules [43].

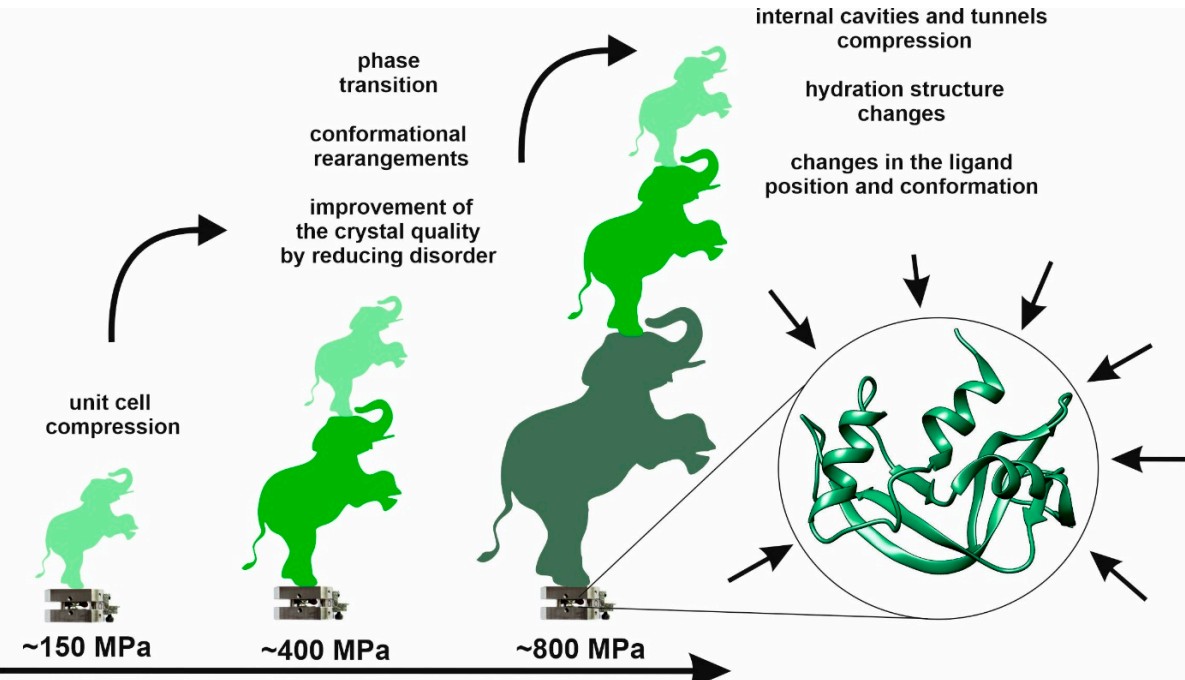

**Figure 2.** Features that can change upon pressurization of the macromolecular crystal.

As observed for small organic molecules pressure turned out to be ideally suited also to study the physical behavior of compressed macromolecules. Since weak intermolecular interactions are most likely to be easily compressed and conformations of molecules themselves can be modified it can lead to ordering and structural phase transition [37].

The application of hydrostatic pressure offers a genuine opportunity to manipulate and explore the intermolecular "bonding" present in crystals. For example, the ordering effect of pressure on the crystal lattice was observed for P23 cowpea mosaic virus (CPMV) crystals [36]. In addition, subjecting the crystals to a pressure of 350 MPa resulted in a significant improvement in diffraction, as the crystals underwent a phase transition from an apparently primitive cell to the body-centered I23 space group. Modest pressures significantly enhanced the diffraction quality as the crystal undergoes a phase transition to a space group with enhanced order.

In general, under relatively low pressure (up to 150–200 MPa), due to the high compressibility of protein crystals, compression typically results in a reduction in the unit cell parameters and volume of a few percent [44]. Pressures exceeding 400 MPa can lead to a contraction that is twice as great, highlighting the immense potency of pressure as a thermodynamic variable capable of inducing structural modifications in proteins, without causing denaturation. The reported compressibility of the unit cell volume $\beta v$ for protein crystals is about 170 MPa$^{-1}$ at moderate pressure, whereas the compressibility at high pressure (limit determined by loss of diffraction), as reported by Fourme et al. for hen egg-white lysozyme (HEWL) based on crystal structures at pressure as high as 1 GPa was equal 150 MPa$^{-1}$ [32]. In general, the compressibility of the unit cell volume calculated for macromolecular crystals decreases as pressure increases. It should be underlined that the HPMX also allows the determination of the isothermal compressibility of a protein molecule itself ($\beta_M$), for example the value of tetragonal HEWL lysozyme is 47 MPa$^{-1}$ based on X-ray crystallographic data obtained at 0.1 and 100 MPa [45] and for IPMDH from the nonpiezophile *S. oneidensis* MR-1 (SoIPMDH) dimer is 54 MPa$^{-1}$ [46]. Furthermore, the analyzed structure revealed that contraction of the molecules in most cases is anisotropic and decreases with increasing pressure. It is worth to mention that with the change of the pressure, the solvent-accessible volume may change, and the solvent molecules can be transferred both sides between pool and the crystals, according to the thermodynamic equilibrium, as it was demonstrated for the CuZnSOD crystals [47].

Analysis of the B-factors can be used as an indicator of the relative vibrational motion of different parts of the structure. Therefore, atoms with low B-factors can be considered as parts of the structure that are well ordered, while atoms with higher B-factors are identified in parts of the structure that are more flexible. Colloc'h et al. in studies on neuroglobin high pressure structures revealed that the increase in the B-factors proved the destabilization of the zone close to the heme, whereas the zone at the back of the protein is stabilized by pressure which was accompanied with a decrease in the B-factors [48]. Furthermore, the general observation of the behavior of secondary structure upon pressurization revealed that beta-sheets are noticeably less deformed than helices.

As aforementioned, the way water interacts with the biomolecules is also rearranged as a result of pressurization. A good example is the analysis of changes in the hydration structure in lysozyme crystals described in [38]. When high pressure is applied, the conformation of the amino acid side chains on the surface of the protein changes and stabilizes the hydrogen bond network, causing more water molecules to be visible. After the pressure exceeded a certain limit and the protein started to unfold, the number of ordered water molecules in the structure decreased.

As pressure is applied, the protein cavities typically are monotonically compressed; however, it was also observed that the volume of a cavity located at the dimer interface can increase, as investigated, for example, in the structure of SoIPMDH where parallel to this volume increase, changes in the hydration shell and water penetration into the cavity were observed [46]. It is also possible that some cavities can shrink sufficiently enough to be undetectable using the 1.2 A radius solvent probe as the pressure increases [49]. Another possible scenario after protein compression describes the generation of a new cleft or tunnels (favorably on the molecular surface) accompanied by water penetration [40]. Such water-penetration phenomena are considered to be initial steps in the pressure-denaturation process. The example of such an event was presented by Hamajima et al. [50], where the

authors showed the decrease in DHFR activity under elevated pressure and connected it with the hydrophobic cavity penetration by three water molecules. It was concluded that increasing number of water molecules entering the spaces between subunits of oligomeric proteins can lead to the disruption of quaternary structure, since it is stabilized mainly by hydrophobic forces [51].

As mentioned above, pressures below 200 MPa typically cause only minor disturbances to the overall structure of proteins, resulting in atomic displacements of only a few tenths of an angstrom. At higher pressure shifts of atoms or structural regions are larger, but still can not to be compared in scale with, for example, loops rearrangements or conformational changes observed upon ligand binding. Even though structural perturbations are relatively small, it does not mean that the functional effects are also small. As exemplified in high pressure studies of urate oxidase (UOX), compression of a protein–ligand complex drove the thermodynamic equilibrium towards ligand saturation of the complex and revealed a new binding site [52]. Furthermore, the UOX after decompression displayed a pressure-dependent decrease in specific activity that culminated at 200 MPa when the complete loss of activity was observed [53]. This may be associated with a disruption of the tertiary structure, usually occurring above 200 MPa. At this pressure, internal amino acid residues could be exposed, leading to the disruption of tertiary structure and increasing the surface hydrophobicity [54]. Further increasing of pressure could lead to the breakage of disulfide bonds [55] and rearrangement of the intermolecular hydrogen bonds [56]. Intermolecular hydrogen bonds are usually broken at lower pressures [57], which is sometimes compensated by new intramolecular interactions [58].

Accordingly, the presented examples prove that high-pressure perturbation potentially enables the trapping in crystal states the protein conformations of biological significance. It is worth mentioning that, for different proteins, the level of pressure required for capturing those conformations can greatly vary and cannot be easily predicted, mostly because the architecture of macromolecules may be less and more resistant to external factors.

## 3. High Pressure Macromolecular Crystallography Instrumentation

The method uses a diamond anvil cell especially configured to maintain protein crystals at a precisely determined pressure. The most widely utilized design for high pressure cells is that of the diamond-anvil cell. The creation of the first DAC chamber dates back to the second half of the 1950s, when it was used for HP IR measurements [59]. A few years later, special DAC cells for powder diffraction [60] and single-crystal X-ray measurements [61] were developed. Placing the metal gasket in between two diamond culets allowed to measure the influence of the applied pressure on the freezing properties of many liquids, but also gave rise to the study of crystals surrounded by hydrostatic medium. Today, DACs are commercially available in various shapes and sizes and can be easily adjusted to different experimental techniques, such as optical spectroscopy (IR, fluorescence, RAMAN), X-ray, magnetic, or electrical measurements [62]. The most popular version of this compact device (ca. $4.0 \times 3.2 \times 1.9$ cm) which can generate pressures up to 10 GPa, and can be effortlessly integrated into a standard diffractometer setup. The essential components of the DAC are shown in Figure 3. Diamonds used in DACs are usually gem quality single crystals, with the carat weight between 1/8 to 1. They are cut and polished along certain crystallographic planes to obtain the brilliant. The culet is usually flat, but other designs such as bevels and toroids are also available [63,64]. To subject the crystal to high pressure diamond cell, it is placed in the sample chamber, a hole drilled in a metal gasket held between the diamond anvils. Pressure is applied to the support plates by tightening screws, which is transferred via the anvils and a hydrostatic medium to the small crystal. In the past, the diamond anvils were supported on beryllium disks. While beryllium is transparent to short-wavelength X-rays, it can introduce a pervasive background that contaminates the diffraction pattern and causes various issues. Therefore, alternative supports are currently being developed. In modern DACs, depending on

the application, the supports are made predominantly from steel, tungsten, titanium, or metal alloys.

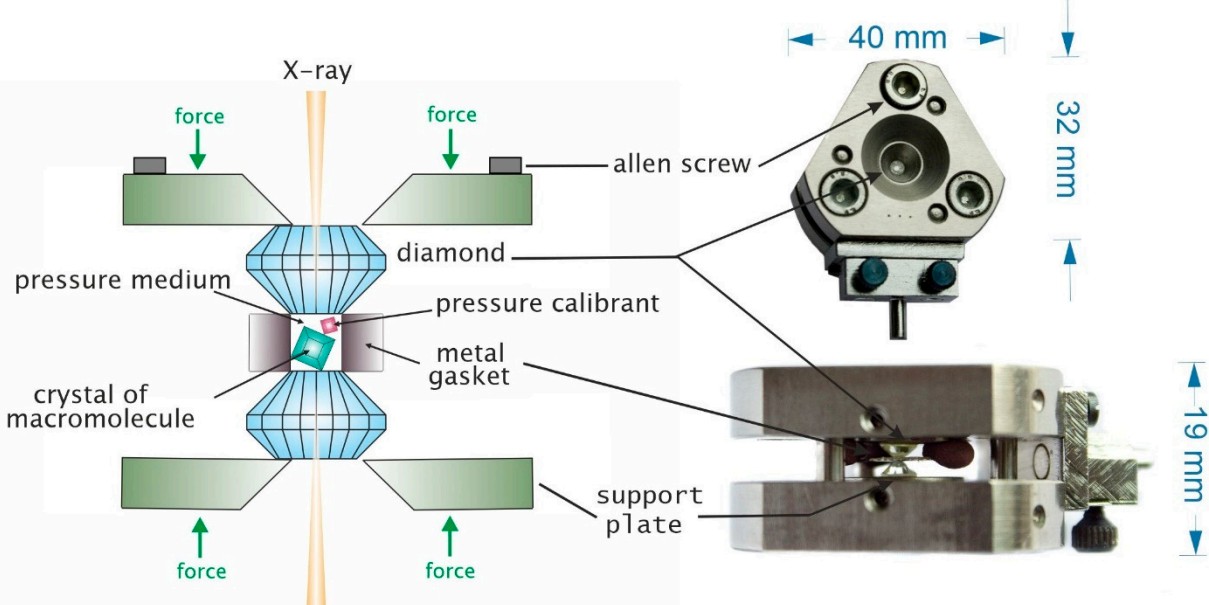

**Figure 3.** Diamond anvil cell: scheme of the core body of the 60-degree diamond anvil cell (on the **left**), a photograph of the core body of the cell once assembled, before it is put in position on the goniometer (on the **right**). The drawing is scaled for clarity.

Pressures inside the DAC are usually measured by placing a small piece of a calibrant (e.g., ruby) within the chamber and gauging the pressure-dependent shift in its fluorescence band. For the first time ruby was used to determine the pressure in the cell in the 1970s and it revolutionized the field [65]. The procedure requires only a standard Raman or other spectrometer and is non-invasive. Despite the fact that DACs have a relatively simple structure, their use in biocrystallographic studies requires several obstacles to overcome. First, it is necessary to mention that there is no one-size-fits-all strategy that can be used to measure all macromolecular crystals; however, a general workflow of the HPMX experiment can be identified (Figure 4).

As already mentioned, macromolecules are complex three-dimensional structures that can be affected to varying degrees by pressure, so the pressure range over which they are studied cannot be standardized. Moreover, from an experimental point of view, there are many factors to consider when planning HPMX measurements. The first challenge, especially at lower pressures and single-crystal measurements, is prevention of crystal movements. Since the DAC is rotated during the measurements, it is important to immobilize the crystals prior to data collection. That can be achieved by adding MPD or PEG in high concentration (20–30%) to the mother liquor [66], but also by placing low absorbing materials, such as cotton fibers [67] or cigarette filter fibers [46] inside the DAC. The crystal space group and its orientation inside the cell are also crucial for the success of the measurements. Low-symmetry crystals require more frames to be collected and are hard to achieve, especially with anisotropic crystals. To overcome this, splinters made from various materials, such as crushed diamonds or boron nitride, can be closed inside DAC together with the sample crystal [68]. Another factor that significantly influences the result of the HPMX experiment is the pressure medium. Although crystallization mother liquor is a natural choice, one should foresee two effects that can occur during sample loading. First, because a very small volume of the mother liquor is used, the most volatile components can evaporate, which may change the composition of the hydrostatic medium and result in destabilization of the crystal before pressure ramping. Second, in

case of high concentration of the precipitant, i.e., salt, formation of additional crystals can be observed upon pressurization, as reported by Kurpiewska et al. in studies of RNase A crystallized from the solution containing a high concentration of ammonium sulphate [49]. As a consequence, some undesirable strong reflections can be present on the diffractograms and disturb data processing.

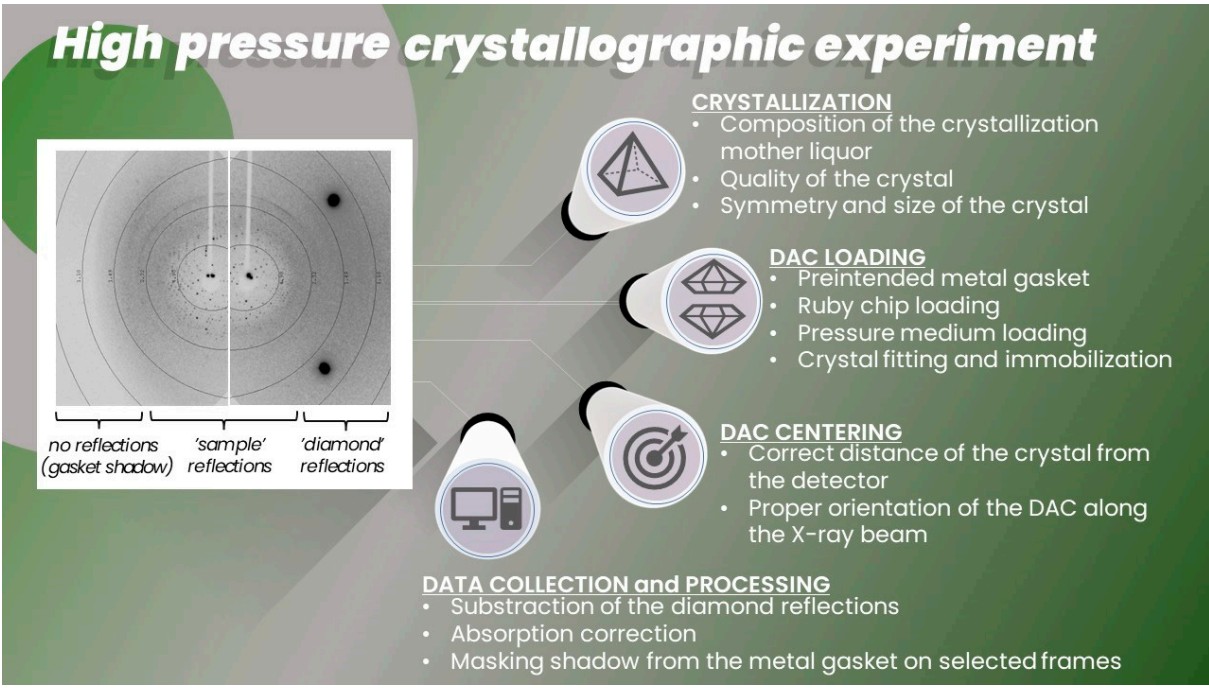

**Figure 4.** Schematic representation of a HPMX experiment.

The limit of pressure stability of the crystals is specific to each study and depends on many elements, i.e., composition of the mother liquor, native or mutated form of protein, or complexation with ligand. Loss of diffraction caused by a loss of long-range order in the crystal was observed near 820 MPa for HEWL, at 400 MPa for (CpMV) [69], while crystals of RNase A, as well as those formed by Cu/Zn superoxide dismutase molecules and DNA fragments could be compressed without loss of diffraction beyond 1 GPa and 2 GPa, respectively [43,49,70]. On the opposite side of the pressure sensitivity scale one can find urate oxidase crystals, for which the critical pressure was determined at about 180 MPa [71]. Regardless of pressure limit for the studied system, a high-pressure experiment should commence with the acquisition of a dataset at ambient pressure. This step is primarily taken to ensure that there are no issues with the experimental setup. Subsequently, additional datasets are collected following the gradual application of pressure. The number of data sets collected is determined by the original objective of the experiment, the compression behavior exhibited by the sample during the course of the experiment, and the point at which failure of the crystal or the metal gasket is observed. A preliminary survey involving large increments in pressure may highlight regions of interest that can then be further investigated in detail.

Even though HPMX experiments can be easily carried out by utilization of DAC and in-house diffractometer, synchrotron centers by enabling the tunable wavelengths and adjustable size of the beam significantly facilitate HPMX. There are many synchrotron beamlines that allow DACs to mount, but only a few are suitable for high pressure measurements for biological samples. The main limitation is the weight of the DAC cells. Standard diffractometers and other positioning units are equipped with very precise motors to adjust the sample position. Such motors are not designed to handle heavy loads. Another difficulty is the available space at the sample environment. The DAC, even in its smallest version, is incomparably greater than the sample holder used for regular crystallographic

experiment. In many beamlines, due to the proximity of cooling attachments, cameras, and other goniostat components, DAC cell cannot be safely mounted. Nevertheless, some beamlines overcame this by mounting modular diffractometers [72] or by disassembling the attachments surrounding the sample. Examples of such beamlines are BL2S1 at the Aichi Synchrotron (Japan), ID7B2 at CHESS (USA), ID09 and ID27 at ESRF (France), CRISTAL at SOLEIL (France), NW12A of the Photon Factory (Japan), I19 at DIAMOND (UK), and beamline under construction at SOLARIS (Poland). The DACs can be mounted on standard goniometer heads or on specially developed grippers (Figure 5).

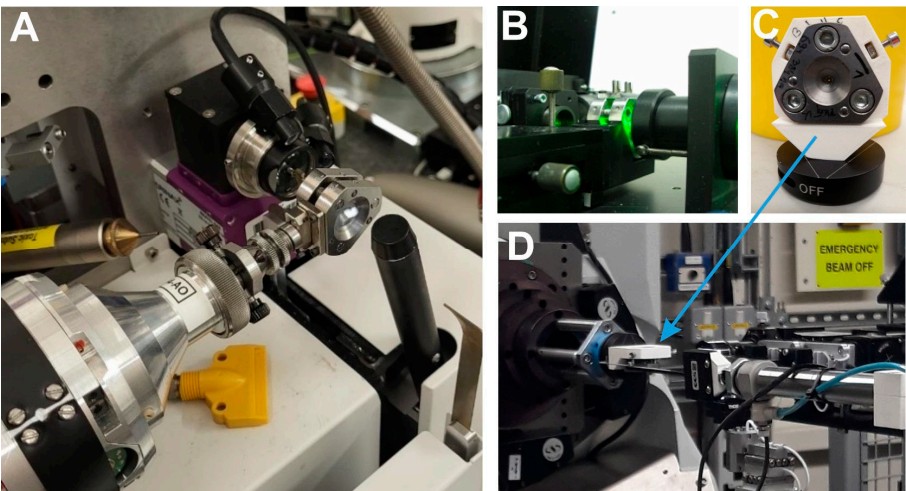

**Figure 5.** High pressure X-ray experiment: (**A**) mounting DAC with the use of goniometer head (ESRF, beamline ID30B, France); (**B**) determination of the pressure inside the DAC (green light of the laser is used to excite the ruby fluorescence and the proportional shift of line R1 694.25 nm determines the pressure inside the DAC); (**C**) the holder produced with 3D printer; (**D**) mounting DAC with the fitted holder (DIAMOND, beamline I19, UK).

Availability of the new generation synchrotron sources allowed to change the experimental approach by allowing a different data collection strategy. With a smaller beam, it is possible to collect data from multiple crystals loaded to the DAC simultaneously. If the crystals packed in the DAC are oriented differently, by collecting several data series in different places, it is possible to obtain a complete diffraction data set even for crystals with lower symmetry. Moreover, packing several crystals at the same time significantly reduces the sample preparation time prior to the experiment. The problem in this approach is primarily to find the right solution that will prevent the crystals from moving after applying pressure, while not affecting the stability and quality of the crystals. Moreover, it may be difficult to orient crystals in the chamber, especially in the case of anisotropic crystals.

In recent years, interest in measurements of biological samples at high pressures has increased. This is reflected in the development of new techniques for pressure measurements different from DAC implementation. One of them is the technique of freezing crystals under pressure. Crystals are harvested and placed in a drop tubes and pressurized with compressed gas up to 200 MPa prior the flash-cooling in liquid nitrogen [73]. This technique allows to reach pressures much lower than inside the DACs, but does not require access to any specific beamline: crystals are mounted in standard pins compatible with most MX beamlines and home source diffractometers.

## 4. Examples of HPMX Achievements and Applications

We will refrain from listing the results or systematically reviewing the extensive literature related to the utilization of high pressure in structural analyses of macromolecules. As indicated by the graph below, a large number of studies have been reported in the last 25 years (Figure 6).

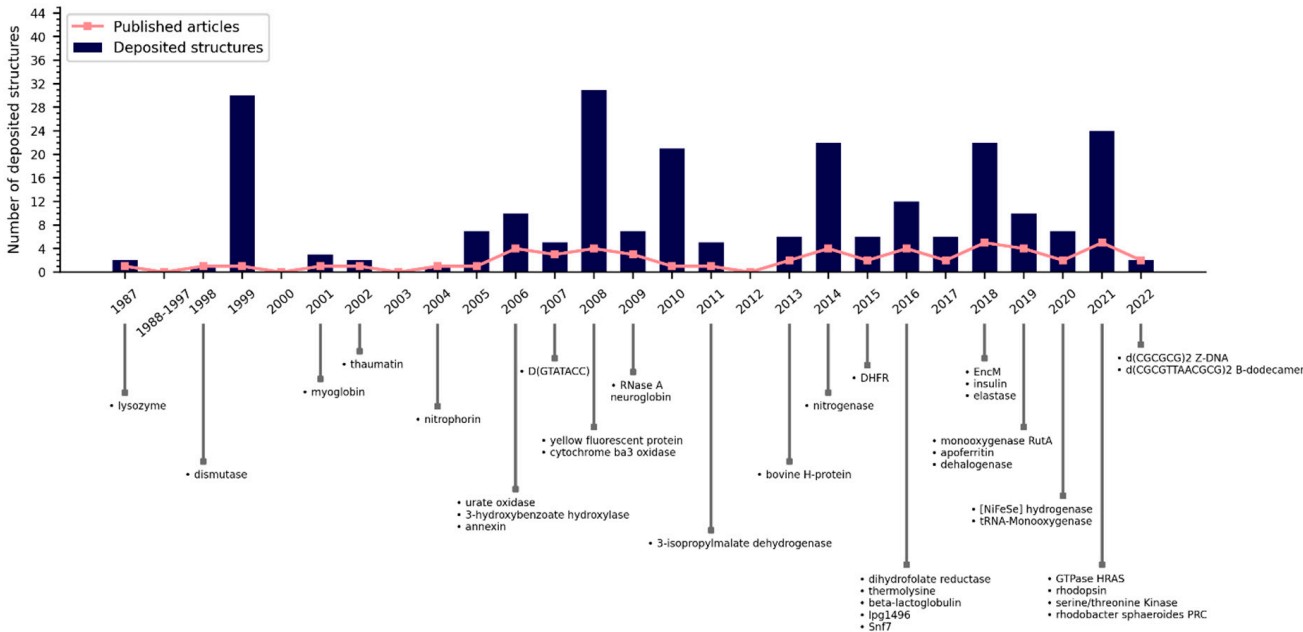

**Figure 6.** A timeline of high-pressure structural studies reflected in the number of depositions in Protein Data Bank. The names of macromolecules appear next to the year of the first HPMX studies focused on the exact biomolecule.

Our previous review [74], and a further analysis of HPMX achievements in 'High Pressure Bioscience, Basic Concepts, Applications and Frontiers' by Wakanabe [75] contain expositions and references that would be redundant to this review. Thus, we rather concentrate on the results of recent high-pressure research and on the discussion of the present state of HPMX. Nevertheless, we decided to list all achievements in high pressure studies of biomacromolecules (Table 1) after 2010 (our last review of PDB content) in order to illustrate the progress in the field.

**Table 1.** High pressure crystal structures of macromolecules deposited to PDB after 2010.

| Sample | Space Group | Pressure [MPa] | Resolution [Å] | PDB id | Reference | Structure of Monomer |
|---|---|---|---|---|---|---|
| Cu,Zn Superoxide dismutase | P2₁2₁2₁ | 570 | 2.00 | 3HW7 | [43] | |
| Urate oxidase | I222 | 0.1; 0.2; 0.5; 0.8; 1.0; 1.5; 1.8; 2.0; 3.0; 3.2; 3.5; 4.2; 6.5; 9.0 | 1.10–1.85 | 3PK5, PK6, 3PKF, 3PK8, 3PLE, 3PKK, 3PKL, 3PKU, 3PLG, 3PJK, 3PKH, 3PKS, 3PLH, 3PLI, 3PKG, 3PKT, 3PLM, 3PLJ, 3PK3, 3PK4, 5FRC, 6I9X, 6I9Z, 6IC1 | [52,76,77] | |
| | | 12; 22 | 1.69–2.36 | 6IA1, 6IA1, 6I19 | [52] | |
| | | 200; 210; 310 | 1.50–2.40 | 6RGM, 7P0D, 7P0C | [52] | |
| | P2₁2₁2 | 60; 100; 150; 200 | 1.64–2.19 | 6IA9, 7PUF, 7PWN, 7Q09 | [52] | |
| 3-Isopropylmalate dehydrogenase from *Shewanella oneidensis MR-1* | C121 | 0.1 | 1.90 | 3WZV, 3WZX | [50] | |
| | | 160, 340, 410, 580, 650 | 1.55–2.20 | 3VL2, 3VL3, 3VL4, 3VL6, 3VL7, 3WZY, 3WZW | [46,50] | |

**Table 1.** *Cont.*

| Sample | Space Group | Pressure [MPa] | Resolution [A] | PDB id | Reference | Structure of Monomer |
|---|---|---|---|---|---|---|
| Lysozyme | P4₃2₁2 | 0.1; 3; 4.5; 10.0 | 1.20–1.65 | 4NWE, 4NWH, 4WLD, 5FSJ, 5FST | [38,77,78] |  |
| | | 190, 280, 380, 440, 500, 600, 710, 800, 890, 920 | 1.55–1.70 | 4WLT, 4WLX, 4WLY, 4WM1, 4WM2, 4WM3, 4WM4, 4WM5, 4XEN, 6Q8R *, 8F2G * | [38] | |
| | P4₃ | 950 | 1.85 | 4WM6 | [38] | |
| Myoglobin | P12₁1 | 3 | 1.55–1.60 | 4NXA, 4NXC | [78] |  |
| Murine neuroglobine | H32 | 1; 2; 3; 5; 8; 10; 15 | 1.60–2.31 | 4O4T, 4O4Z, 5MJD, 5MJC, 5NVI, 5NW6, 5O1K, 5O17, 5O27, 3GK9, 3GKT, 3GKN, 6R1Q * | [78–81] |  |
| | | 150, 240, 270, 280, 310 | 1.90–2.40 | 5EV5, 5EYJ, 5EOH, 5F0B, 5EQM, 5F2A | [48] | |
| Nitrogenase MoFe Protein from *Clostridium* | P1 | 24.5 | 1.90 | 4WN9 | [82] |  |
| Nitrogenase MoFe Protein from *Azotobacter vinelandii* | P12₁1 | 24.5 | 2.00 | 4WNA | [81] |  |
| Dihydrofolate reductase from *Escherichia coli* complexed with folate and NADP+ | P12₁1 | 0.1 | 1.70 | 4X5F | [83] |  |
| | P12₁1 | 270 | 1.90 | 4X5G | [83] | |
| | C121 | 500, 660, 750 | 1.80–1.90 | 4X5H, 4X5I, 4X5J | [83] | |
| Dihydrofolate reductase from *Escherichia coli* | P2₁2₁2₁ | 0.1 | 1.80 | 5Z6F, 5Z6L, 5Z6M, 5Z6J, 5Z6K | [83] |  |
| Thermolysin | P6₁22 | 4; 10; 45 | 1.20–1.70 | 5FSP, 5FSJ, 5FSS | [77] |  |
| Bovine beta-lactoglobulin complex with dodecane | P3₂21 | 430-550 | 2.85 | 5IO7, 5LKF | [84] |  |
| Uncharacterized Protein Lpg1496 from *Legionella pneumophila* | P4₂2₁2 | 300; 350 | 1.48 | 5T8B *, 5T8C * | - |  |
| Vacuolar-sorting protein Snf7 | P12₁1 | 200; 350 | 2.20 | 5T8N *, 5T8L * | - |  |

**Table 1.** *Cont.*

| Sample | Space Group | Pressure [MPa] | Resolution [A] | PDB id | Reference | Structure of Monomer |
|--------|-------------|----------------|----------------|--------|-----------|----------------------|
| Putative FAD-dependent oxygenase EncM complexed with dioxygen | C222 | 0.5; 1; 1.5 | 1.39–2.04 | 6FOW, 6FOQ, 6FP3, 6FYC | [85] | |
| Putative FAD-dependent oxygenase EncM complex with xenon | I222 | 1.5 | 3.65 | 6FY9 | [85] | |
| Dihydroorotase from *Aquifex aeolicus* | C222$_1$ | 60; 120 | 2.50–2.60 | 6GDF, 6GDD | [42] | |
| Bovine insulin | I2$_1$3 | 60; 100; 200 | 2.00–2.15 | 6Q8Q *, 6QQG, 6QRK, 6QRH | [39] | |
| Elastase (PPE) | P2$_1$2$_1$2$_1$ | 200 | 1.80 | 6Q8S * | - | |
| Thermolysine | P6$_1$22 | 200 | 1.85 | 6QAR * | - | |
| Horse spleen apoferritin | F432 | 200 | 2.00 | 6RA8 * | - | |
| Monooxygenase RutA complexed with dioxygen/uracil and dioxygen | P3$_1$21 | 0.5; 15 | 1.80 | 6SGG, 6TEF, 6TEEG | [86] | |
| Hyperstable haloalkane dehalogenase variant DhaA115 | P2$_1$2$_1$2$_1$ | 15 | 1.55 | 6SP8 | [87] | |
| [NiFeSe] hydrogenase from *Desulfovibrio vulgaris* hildenborough | P12$_1$1 | 7.5; 10 | 1.09–2.20 | 6Z7R, 6Z8J, 6Z8O, 6Z9G | [88] | |
| | C121 | 7.5; 10 | 1.02–1.53 | 6Z8M, 6Z9O | [88] | |
| tRNA-monooxygenase | C121 | 200 | 1.80–2.50 | 6ZMC, 6ZMA | [89] | |
| GTPase HRAS | P3$_2$21 | 200; 500; 650; 900 | 1.70–2.10 | 7OGE, 7OGF, 7OGA, 7OGC, 7OGB | [90] | |

**Table 1.** *Cont.*

| Sample | Space Group | Pressure [MPa] | Resolution [A] | PDB id | Reference | Structure of Monomer |
|---|---|---|---|---|---|---|
| Proton pump MAR rhodopsin | P121 | 13.0 | 2.25 | 7Q37 | [91] |  |
| Mutant bacterio-rhodopsin | C222 | 13.3; 13.2 | 2.00 | 7Q35, 7Q38 | [91] |  |
| KR2 sodium pump rhodopsin | I222 | 13.1 | 2.60 | 7Q36 | [91] |  |
| RNase A | $P3_221$ | 22; 50; 72; 100; 120 | 1.48–1.60 | 7Q7A *, 7Q77 *, 7Q76 *, 7Q79 *, 7Q78 * | - |  |
| Human Serine/Threonine kinase | $P4_122$ | 111 | 2.60 | 7Q7D * | - |  |
| *Rhodobacter sphaeroides* Photosynthetic Reaction Center | $P4_122$ | 75; 100 | 2.55–2.87 | 7Q7M *, 7Q7J *, 7Q7G *, 7Q7N * | - |  |
| d(CGCGCG)2 Z-DNA (Rich's DNA) | $P2_12_12_1$ | 300; 540; 715 | 1.41–1.44 | 7ZQO, 7ZQN, 7ZQM | [66] |  |
| d(CGCGTTAACGCG)2 B-DNA (Dickerson's DNA) | $P2_12_12_1$ | 310 | 2.55 | 7ZQL | [66] |  |

* Unpublished results.

We have chosen a handful of recently published papers to showcase the state of the art of HPMX and its effective use in conjunction with complementary methods. Current developments demonstrate the impressive use of high macromolecular crystallography to examine the response of macromolecules to changes upon pressurization. In 2018, Nagae [83] and co-workers published results of structural studies that describe the high pressure structure of *Escherichia coli* dihydrofolate reductase complexed with folate and NADP+. The measurements were taken at the Aichi Synchrotron Radiation Center at Nagoya University beamline BL2S1 (Japan). They were carried out in 800 MPa with a resolution up to 1.7 Å. This allowed the production of a complete data set with several crystals. They were placed in the DAC chamber with tied cigarette filter fibers for stability. Analysis of determined structures demonstrated the very successful application of HPMX as a method to study protein dynamics [2,83,92]. At the molecular level, Nagae et al. identified the impact of increasing pressure on three functional loops of the investigated enzyme. The observed changes (opening and closing of loop M20, which plays a significant role in the binding and release of ligands) were followed by a symmetry change from $P2_1$ to C2. Furthermore, the reported findings have also proved that the HP experiment allowed

the high-energy substrates and transient structures related to the protein reaction cycle. Additionally, the study confirmed that compression increases the number of observed water molecules. Finally, it was noted that with increasing pressure, the nature of the direct interaction between the enzyme and the substrate changed to a water-mediated interaction. The measurements are the first example of using the DAC pressure chamber for low symmetry measurements [83].

Food science may be one more potential application of HPMX. Structural measurements at high pressure were used in allergen research in reference to the development of hypoallergenic food products. High-pressure conditions have indicated structural rearrangements that explain the observed changes in the antigenicity of beta-lactoglobulin after pressurization [93]. Kurpiewska et al., using a standard setup of the single crystal closed inside the DAC, established a structure for the bovine beta-lactoglobulin complex with dodecane at pressure up to 550 MPa with a resolution of 2.85 Å using a home-source diffractometer. The high-pressure structures caused reorientation of the subunits by affecting the hydrogen bonds responsible for dimer stabilization. As in the previous case, it was observed that water penetrated the cavities of the protein, even including hydrophobic spaces. Although the protein core remained unchanged at a pressure greater than 500 MPa, changes in the position of amino acid residues and the overall shape of the binding pocket were observed as soon as pressure reached 400 MPa. These results showed that high-pressure macromolecular crystallography can potentially identify the most pressure-sensitive fragments in protein allergens that are responsible for the allergic response.

High-pressure crystallography helped determine the structural and dynamic behavior of the Ras oncogene protein and complements previous research on this topic [94]. Again, by using the DAC, Girard et al. determined the structure of the investigated protein at pressure of 900 MPa with a resolution up to 1.70 Å at the ESRF synchrotron (Grenoble, France) on the ID09 and ID27 beamlines. Only one crystal was used for each data set and translated in the beam every 10° of rotation. Measurements were complemented by NMR spectroscopy at high pressure. This combination of two comprehensive methods revealed compelling changes at the molecular level with high resolutions. Despite the increased pressure no changes have occurred in the P-loop, indicating that the intrinsic rigidity of these regions is directly connected to the Ras cycling efficiency. Another interesting observation refers to the fact that pressure appeared to imitate the allosteric effect. These unexpected results of pressurization open the possibility to explore the structural determinants of proteins without any mutations or introduction of exogenous partners into the system. Further investigations may enable a precise description of Ras sub-states, and a more general view of G proteins sub-states. In terms of future medical application, the reported results encourage the design of inhibitors specifically targeting the low populated conformers of Ras [90].

The last example presented in this review concerns DNA, which is known to be a more pressure-resistant material than proteins. In research published by Prange and co-workers, nucleic acid structures were studied by compression of single crystals inside the DAC [70]. Full data recordings were performed at the ESRF synchrotron ID27 beamline. With the strategy of crystals translated in the beam every 15° of rotation, taking advantage of the small cross-section of the X-ray beam to irradiate successively fresh portions of the sample. This approach has resulted in the limiting of the independent reflections, which were recorded to about 75–80%. As a result, B-DNA (Dickerson's DNA) up to 310 MPa and Z-DNA (Rich's DNA) up to 715 MPa with a resolution of 2.55 Å and up to 1.41 Å were measured, respectively. In the case of Z-DNA crystals, the diffraction pattern was observed even at pressure up to 1.2 GPa [66]. The results obtained complement the information on A-DNA structure and show that the base compression was higher for Z-DNA (320 MPa) and B-DNA (420 MPa) than for A-DNA (130 MPa) [70]. The structures described above are the only DNA structures measured so far in high pressure.

The ability to pressurize biological samples of any shape and size further extends the scope for studies of the crystal state. In the last decade, 20 unique proteins were investigated

under elevated pressure. This shows the growing interest in this type of research, but also shows that there is still much to be discovered.

**5. Perspectives of HPMX**

When Kundrot and Richards published the first protein structure determined under high pressure [95], they started a research field that is still active today. Scientists combine fundamental knowledge of biology with the results of HPMX and other biophysical methods in order to gain a new perspective. We believe that one single method is unlikely to unravel the fundamentals of research fields regarding novel protein design, protein folding pathways, synthetic metabolic routes, as well as protein-aggregation mechanisms, pathogenesis of protein misfolding and disease, and proteostasis networks. It is worth here noting that an important supplement to studies on the behavior of protein structure at high pressures are studies using the small-angle scattering technique (HP BioSAXS) in solution. The BioSAXS technique is complementary to classical protein crystallography and allows verification of the general conformation of protein molecules in solution, where they are not limited by restrictions related to packing in the crystal lattice [96]. In recent years, these studies have found a wider group of users, even though, they are limited by the maximum hydrostatic pressure available in these experiments, typically not exceeding 0.7 GPa [97–99]. However, an extremely attractive prospect for monitoring the behavior of proteins in solution at high pressures is the high-pressure chromatography-coupled small-angle X-ray scattering system combined with synchrotron radiation [100].

Through this review, we have showcased the status of structural analyses conducted under high pressure and demonstrated that this method takes us one step closer to addressing some critical research queries, such as: What is the actual structure–function relationship that governs the pressure stability of proteins? How does high pressure impact the stability and activity of macromolecules? Can the conformational changes triggered by high pressure reduce or enhance the allergenicity of particular proteins? How does HP affect plasticity involved in allosteric equilibria between conformers? Considering that modern HPMX allows to investigate simple and small entities including amino acids [101], DNA fragments [70], medium in size molecules as monomeric proteins, as well as large and complex systems as multimeric proteins and viruses [61], we can expect that answers to more demanding and intriguing questions will be soon received. Mentioning just a few: how are metabolic pathways or networks affected by HP, and can this method be used to improve the yield or performance of industrial microorganisms (not only enzymes)? Additionally, high pressure studies can shed light on the conformational changes, and therefore can help to explore the potential implementation of high-pressure processing for treatment of conformational diseases? Another question worth exploring is: can we predict conditions necessary for HP inactivation of microorganism? Finally, regarding human exterritorial ambitions, we hope to reveal whether it is possible to habitat extreme environments beyond Earth?

**Author Contributions:** Conceptualization, K.K. and J.S.; writing—original draft preparation, K.K., J.S., A.K. and M.K.; writing—review and editing, K.K. and J.S.; visualization, K.K. and A.K.; supervision, M.K. All authors have read and agreed to the published version of the manuscript.

**Funding:** This project was supported by research grant PRELUDIUM BIS (grant ID: 2020/39/O/ST4/03465) from National Science Centre (Poland).

**Data Availability Statement:** Not applicable.

**Conflicts of Interest:** The authors declare no conflict of interest.

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
