# Peer review of "The Promise and Challenge of High Pressure Macromolecular Crystallography"

_crystals, doi:10.3390/cryst13040560_

Round 1
Reviewer 1 Report
The manuscript by Kurpiewska et al. on macromolecular x-ray crystallography (HPMX) represents a timely review of a method that may becoming more and more important in future. The reviewer would have appreciated a much more extensive review with more technical details of the method. However, these can be partially found in the cited report by Roger Fourmé. Nevertheless I recommend the review for publication in crystals.
Author Response
Authors would like to thank for the kind review. The main goal of the Review was to present the achievements of the method rather than a desciption of methodology, thus we have focused on the perspectives for the field with just a short, but essential description of the technical acpects.
Reviewer 2 Report
This is a very current and necessary review. In my opinion Authors fulfilled all assumptions they have made and prepared a very nice and easy-to-read/understand review.
Before publishing, I suggest to make a language review one more time, because I found a few typos.
I my opinion, this paper is suitable for publishing in Crystals.
Author Response
The Authors would like to thank for the kind review. We have checked the manuscript and hope it is free from typos now.
Reviewer 3 Report
Kurpiewska and coworkers summarized the recent development of high-pressure crystallography study of the macromolecules and remaining challenges in this field. This review article also gives a detailed introduction of the common used diamond anvil cell and the high pressure crystallography work flow. The work is well written and the listed examples and references are complete. I would like to recommend it for publication after addressing the following minor changes.
1) It can be expected that upon compression, the structures of macromolecules would undergo change in symmetry, distortion, or closing. The authors can have some detailed structural comparisons for few published examples to make readers clearly understand what high pressure chaneg the structures, and how hiph pressure influence the inter-/intra-molecular noncovalent interactions or folding modes. Some examples have already been listed, but more details are encouraged to compare.
Author Response
We would like to thank for the kind review.
ANSWER to "1) It can be expected that upon compression, the structures of macromolecules would undergo change in symmetry, distortion, or closing. The authors can have some detailed structural comparisons for few published examples to make readers clearly understand what high pressure chaneg the structures, and how hiph pressure influence the inter-/intra-molecular noncovalent interactions or folding modes. Some examples have already been listed, but more details are encouraged to compare."
We have introduced new parts concerning the aspects mentioned by the Reviewer and added more examples on how high pressure can influence biological structures.